# Covert shift of attention modulates the value encoding in the orbitofrontal cortex

Yang Xie[1,2], Chechang Nie[1,2], Tianming Yang[1]*

[1]Institute of Neuroscience, Key Laboratory of Primate Neurobiology, CAS Center for Excellence in Brain Science and Intelligence Technology, Shanghai Institutes for Biological Sciences, Chinese Academy of Sciences, Shanghai, China; [2]University of Chinese Academy of Sciences, Beijing, China

**Abstract** During value-based decision making, we often evaluate the value of each option sequentially by shifting our attention, even when the options are presented simultaneously. The orbitofrontal cortex (OFC) has been suggested to encode value during value-based decision making. Yet it is not known how its activity is modulated by attention shifts. We investigated this question by employing a passive viewing task that allowed us to disentangle effects of attention, value, choice and eye movement. We found that the attention modulated OFC activity through a winner-take-all mechanism. When we attracted the monkeys' attention covertly, the OFC neuronal activity reflected the reward value of the newly attended cue. The shift of attention could be explained by a normalization model. Our results strongly argue for the hypothesis that the OFC neuronal activity represents the value of the attended item. They provide important insights toward understanding the OFC's role in value-based decision making.
DOI: https://doi.org/10.7554/eLife.31507.001

## Introduction

Imagine you are standing before a fruit stand and trying to buy some apples. Facing a box of apples, you usually will pick up one apple at a time, evaluate it, decide whether you will have it or leave it, and then move on to the next apple until you have chosen enough. The apples are evaluated and selected sequentially. The decision to pick up a particular apple for scrutiny is often based on its simple visual features, such as color, size, or texture. Apples with desired salient features are more likely to capture your attention in a bottom-up manner and thus guide the decision-making process. Although the attentional modulation of neural activity in the visual cortices has been extensively studied, it is often in a setting where visual information is processed in parallel and attention is distributed across the visual space (*Moran and Desimone, 1985*; *Motter, 1993*; *Treue and Maunsell, 1996*; *McAdams and Maunsell, 1999*). It is not well understood how serial decision making as in the example above is achieved in the brain and what role attention plays during this process.

We focus our study on the orbitofrontal cortex (OFC), which has been shown to play an important role in representing the association between sensory stimuli and reward during value-based decision-making (*Wallis and Miller, 2003*; *Padoa-Schioppa and Assad, 2006*; *Rudebeck et al., 2013b*; *O'Neill and Schultz, 2015*). The OFC receives visual sensory inputs from inferior temporal and perirhinal cortex, as well as from limbic structures including the amygdala and the cingulate cortex, allowing it to have the information for establishing the association between visual information and reward (*Carmichael and Price, 1995b*, *1995a*). Studies have shown that a significant number of OFC neurons encode the reward value associated with sensory stimuli (*Wallis and Miller, 2003*; *Padoa-Schioppa and Assad, 2006*).

OFC neurons are typically reported to be insensitive to stimulus locations (*Wallis and Miller, 2003*; *Kennerley and Wallis, 2009*; *Grattan and Glimcher, 2014*). When options are presented

*For correspondence:
tyang@ion.ac.cn

Competing interests: The authors declare that no competing interests exist.

simultaneously and the eye movements are controlled as in many studies (*Wallis and Miller, 2003*; *Padoa-Schioppa and Assad, 2006*; *Rudebeck et al., 2013a*; *Cai and Padoa-Schioppa, 2014*; *Blanchard et al., 2015*), it is not immediately obvious how to reconcile OFC neurons' insensitivity to stimulus locations and their roles in encoding the value of each option and in decision making. It has been reported that the value encoding in the OFC is affected by gaze location (*McGinty et al., 2016*) and OFC activities alternated between states during decision making (*Rich and Wallis, 2016*), which led to the suggestion that the value information may be processed sequentially, possibly guided by eye movements or overt attention. Yet a direct testing of the hypothesis has been lacking, and it is not known how covert attention may affect the neuronal activity in the OFC.

In the current study, we aim to test the hypothesis that the activity of OFC neurons represents the value of covertly attended stimulus when multiple options are presented simultaneously. Signals for attention, eye movement, reward, and decisions in the brain are often tangled together, which has prevented the field to understand how value representation in the brain may be modulated by attention (*Maunsell, 2004*). We addressed the issue with a passive-viewing task, which allowed us to tease apart the possible interference of eye movement, reward, and decision when studying the attentional modulation of OFC neural responses. In this task, a pair of visual cues were presented while monkeys were fixating. The monkeys did not have to make any choices. They received the reward associated with one of the two cues, randomly selected, at the end of the fixation period. In some trials, we applied a transient visual perturbation to one of the cues to induce a transient shift of attention when monkeys continued to fixate. The perturbation was irrelevant to the reward outcome. Similar manipulations have been shown to affect both monkeys' behavior and neural responses in the lateral intraparietal area that could be attributed to a bottom-up attention mechanism (*Balan and Gottlieb, 2006*). Thus, we expected the manipulation produced similar shifts of attention in our experiments. We recorded single unit activities in the OFC and observed that the OFC neurons encoded only the larger value when two stimuli were presented without visual perturbations. When a visual perturbation was applied, their activity switched to reflect the value of the perturbed stimulus. The attention modulation of OFC neurons can be described with a normalization model of attention shifts.

## Results

### Behavior

Two monkeys were trained to perform the passive-viewing cue-reward association task (*Figure 1AB*). To find out if the monkeys learned the cue-reward associations, we measured the pupil dilation when the monkeys were performing the task. When only one cue was presented, there was no uncertainty in the amount of reward that the monkeys would get. Although the dynamics of pupil responses during the cue presentation period differed between the two monkeys, both monkeys showed pupil dilation patterns clearly indicating they understood the reward association (*Figure 2AB*). For the purpose of this study, we focused our analyses of pupil dilation on the period immediately after the cue presentation. During this period, there were no visual stimuli other than the fixation point on the screen, the pupil responses showed the largest separation between different reward conditions, and similar patterns were seen in both monkeys. A linear regression showed that the pupil size increased for both monkeys during the analysis time window when the reward associated with the cue was larger ($p \ll 0.001$ for both monkeys).

Next, we looked at the pupil dilation pattern in the double-cue conditions. The pupil responses were dominated by the cue that predicted a larger reward. When we pooled the cue combinations in which the cue with the larger reward was the same, the pupil dilation was similar to the condition when the higher value cue was presented alone (*Figure 2CD*, *Figure 2—figure supplement 1*, and *Supplementary file 1*, Table 2). Although a two-way ANOVA analysis showed a significant difference between the single- and double-cue conditions (p=0.0027 for monkey D and 0.0239 for monkey G), the post hoc Tukey test showed no individual pairs of reward value conditions were significantly different (0.9996, 1, 0.7448, 0.9397, and 0.9932 for monkey G and 0.9999, 0.9836, 0.9100, 0.0750, and 0.4317 for monkey G). The results suggested the higher value cue dominated the pupil responses. In these conditions, the computer randomly selected one of the cues at the end of a trial and delivered its associated reward to the monkey. Nevertheless, the pupil dilation did not reflect either the sum

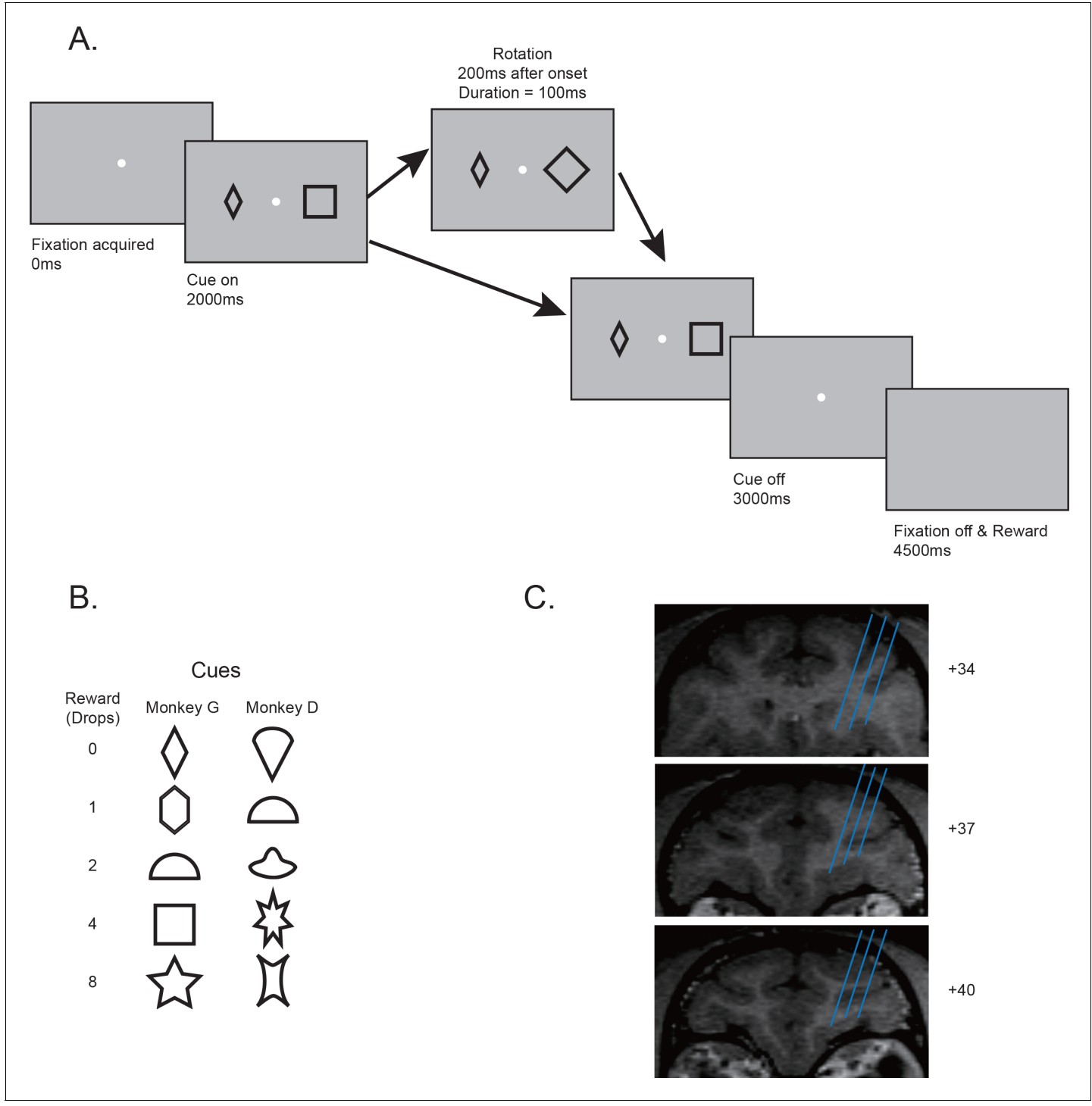

**Figure 1.** Behavior paradigms and electrophysiology recording locations. (**A**) Behavior paradigm. The monkeys had to maintain their fixation while passively viewing one or two visual cues presented on the screen. Each cue was associated with a reward. In some trials, one of the cues was quickly rotated back and forth for 100 ms. At the end of the trial, the monkey received reward associated with one randomly selected cue from the pair. (**B**) Cue-reward associations used in two monkeys. (**C**) Estimated recording locations. The structural MRI images shown were from monkey G.

DOI: https://doi.org/10.7554/eLife.31507.002

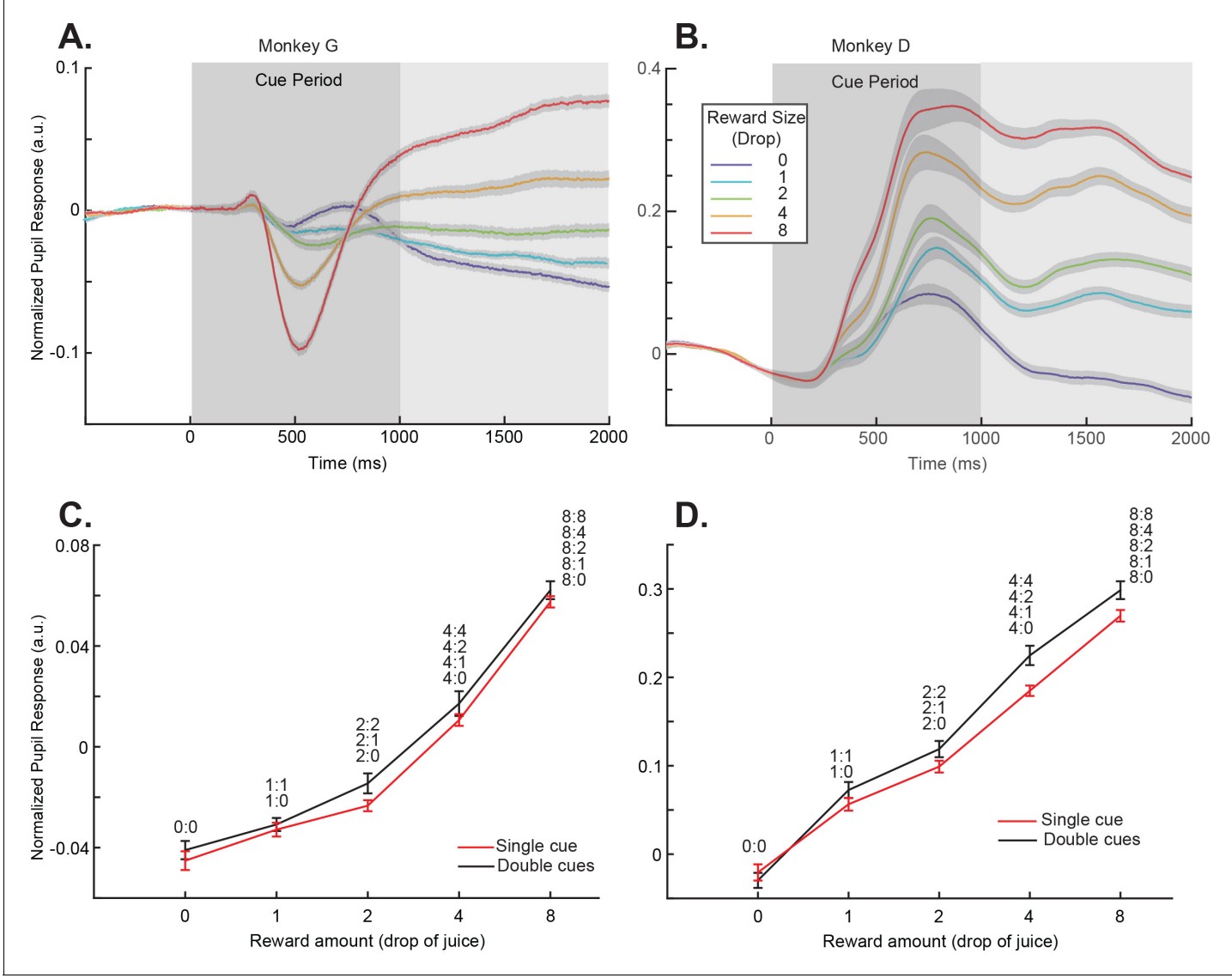

**Figure 2.** Pupil dilation responses reflected cue-reward association. (A) Pupil dilation responses of trials in the single-cue conditions for monkey G. Time 0 indicates the cue onset. The dark gray box indicates the cue presentation period. The light gray box indicates the period in which the mean pupil responses were calculated and plotted in panel (C). The colors indicate cues with different rewards. The shading around each curve represents s. e.m. between sessions. (C) Pupil dilation responses of both the single- and double-cue condition trials for monkey G. The responses are calculated as the average pupil size within 1 s after the cue offset, indicated by the light gray box in panel (A). Trials in the double-cue conditions (black curve) are grouped by the higher value between the two cues. The numbers near each data point indicate the cue combinations that comprise each group. The error bars represent s.e.m. between sessions. (B) and (D) Plots for monkey D.
DOI: https://doi.org/10.7554/eLife.31507.003

The following figure supplements are available for figure 2:

**Figure supplement 1.** Pupil responses for each cue combination.
DOI: https://doi.org/10.7554/eLife.31507.004

**Figure supplement 2.** The pupil responses did not reflect the value of the perturbed cue.
DOI: https://doi.org/10.7554/eLife.31507.005

**Figure supplement 3.** The monkeys' eye positions during fixation were not affected by visual perturbations.
DOI: https://doi.org/10.7554/eLife.31507.006

of the cue values or the expected value, which was the mean of the two. This was consistent with the idea that the monkeys were primarily paying attention to the cue associated with a larger reward, and the pupil dilation reflected its value.

Overall, the pupil dilation responses suggested that the monkeys understood the cue-reward associations under the passive viewing condition. We next explored how the associations were represented in the OFC.

## OFC neural responses without visual perturbation

After verifying that the monkeys understood cue-reward associations used in the task, we went on recording neural activity from the OFC. We recorded from a total of 846 neurons in Walker's areas 11 and 13 (*Walker, 1940*), between the lateral and medial orbital sulci (*Figure 1C*). Among the 846 neurons, we identified 232 neurons that exhibited significant responses during the cue period. Among them, 110 neurons were found to be selective for reward value (*Table 1*).

An example value-selective OFC neuron is shown in *Figure 3*. In the single-cue conditions, its responses were modulated by the value associated with the cue (*Figure 3AB*). Moreover, when we presented two cues together, the neuron's response pattern mimicked the pattern of pupil responses (*Figure 3B* and *Figure 3—figure supplement 1*). The responses to the single- and double-cue conditions were similar (two-way ANOVA, p<1e-7 for reward value, 0.16 for number of cues). A linear regression further revealed that only the higher value cue affected the neuron's responses (p=3.06e-7 for higher value and 0.51 for lower value). The neuron's responses only reflected the value of the higher value cue, which presumably captured the monkeys' attention.

This pattern was true in general for the population of neurons that encoded value in the OFC. We divided these neurons into two groups. The 73 positively tuned neurons showed greater responses when cues with larger rewards were presented, and the 37 negatively tuned neurons showed greater responses when cues with smaller rewards were presented (*Table 1*).

The population average responses of both groups of neurons showed similar patterns as the example neuron (*Figure 3CD*). The two-way ANOVA showed the number of cues did not affect the population responses of each group (p=0.50 for the positively tuned neurons, 0.29 for the negatively tuned neurons). The linear regression of the population response on the values of both cues showed that only the higher value affected the responses significantly (higher value: p<1e-7 for both groups; lower value: p=0.70 for the positively tuned group and 0.14 for the negatively tuned group).

Not surprisingly, each individual neuron, regardless whether categorized as positively tuned or negatively tuned, showed similar responses in the single- and the double-cue conditions, when the double-cue condition trials were grouped by the higher value (*Figure 3E*). The mean responses between the two conditions were not significantly different (two-tailed t-test, p=0.12 and 0.27 for the positively and the negatively tuned neurons).

These results suggested that the value-encoding OFC neurons' responses were highly homogenous and dominated by the higher value cue when multiple cues were presented simultaneously.

**Table 1.** Numbers of OFC neurons recorded and classified in this study.

For neurons with significant attentional modulation, we defined the consistency of the modulation as whether their responses to the double-cue conditions with visual perturbation became more or less similar to their responses to the single-cue condition when the perturbed cue was presented alone.

| | | Value-selective neurons | Positively tuned neurons | Negatively tuned neurons | Visually responsive neurons | Total# neurons recorded |
|---|---|---|---|---|---|---|
| No attentional modulation | | 67 | 44 | 23 | 232 | 846 |
| With attentional modulation | Consistent | 40 | 28 | 12 | | |
| | Inconsistent | 3 | 1 | 2 | | |
| | Total | 43 | 29 | 14 | | |
| Total | | 110 | 73 | 37 | | |

DOI: https://doi.org/10.7554/eLife.31507.007

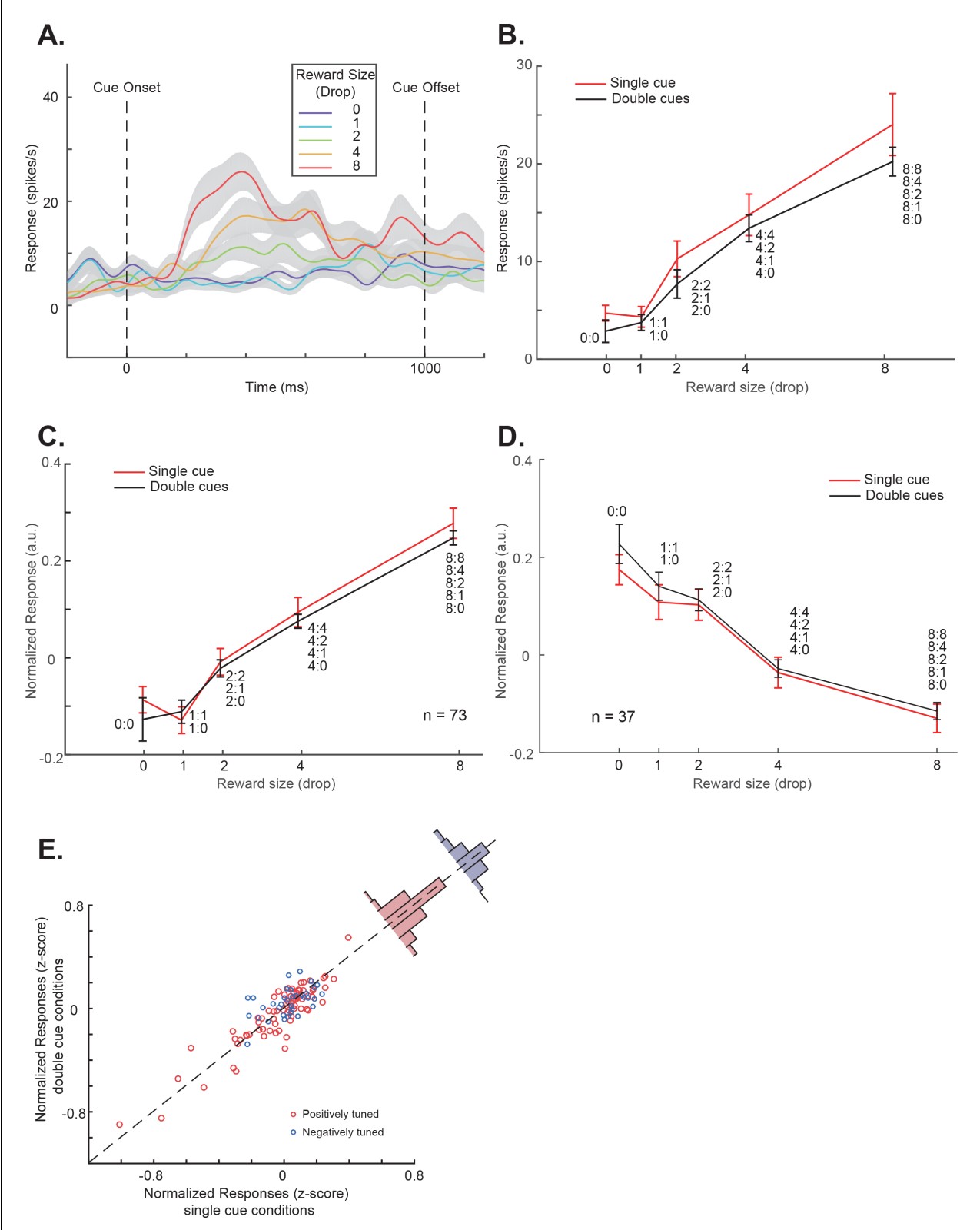

**Figure 3.** OFC responses to cue conditions without perturbations. (**A**) The PSTH in the single-cue condition of an example OFC neuron. The trials are grouped by the cue's associated reward value indicated by different colors. The shading around each curve represents s.e.m. between trials. (**B**) The cue responses to both the single- and double-cue conditions of the example neuron. Trials in the double-cue conditions (black curve) are grouped by the higher value between the two cues. The numbers near each data point indicate the cue combinations that comprise each group. The error bars

*Figure 3 continued on next page*

*Figure 3 continued*

represent s.e.m. between trials. (C) and (D) The population cue responses to both the single- and double-cue conditions of the positively (C) and the negatively tuned (D) OFC neurons, plotted in a similar format as in panel B. The error bars represent s.e.m. between neurons. (E) The responses of each OFC neuron to both the single- and double-cue conditions. Red and blue data points indicate the positively and the negatively tuned neurons. The red and blue histogram insets are the distributions of response differences between the single- and the double-cue conditions for the positively and the negatively tuned neurons, respectively. Double-cue condition trials are grouped by the higher value.
DOI: https://doi.org/10.7554/eLife.31507.008

The following figure supplement is available for figure 3:

**Figure supplement 1.** Firing rates of the example neuron in *Figure 3AB* for each cue combination.
DOI: https://doi.org/10.7554/eLife.31507.009

Note that, for the negatively tuned neurons, the dominating cue was the non-preferred cue. Therefore, their responses were suppressed by the presence of a higher value cue. The results were consistent with the idea that the monkeys were mostly paying attention to the cue associated with the larger reward and the activity of OFC neurons reflected the value of the attended cue.

## OFC neural responses with visual perturbation

So far, we have presented data from the conditions in which the monkeys were just passively looking at a pair of visual cues. We assumed that the monkeys' attention was at the higher value cue between the two. The assumption agrees with the hypothesis that the OFC neurons encode the value of the attended cue. To strengthen the argument, we further introduced visual perturbations that would attract monkeys' attention toward one of the cues in the double-cue conditions.

The perturbation we introduced was a transient rotation of one of the cues that lasted only 100 ms. The perturbation could be applied to either the higher value or the lower value cue. It was independent of the assignment of the rewarded cue, and the monkey would not gain any behavioral advantage, knowledge, or additional reward by paying attention to it. Indeed, we observed no behavior indications that the monkeys were responding to the perturbations. They induced no changes in pupil size, and the monkeys' eyes were not attracted toward the perturbed location (*Figure 2—figure supplements 2* and *3*). It has to be noted that pupil dilation responses are influenced by many sensory and cognitive parameters, and the modulation by value is relatively modest. They also have slow dynamics. The quick rotation we used was subtle and transient. Those facts might obscure potential pupil responses induced by the perturbations.

However, these subtle visual perturbations affected OFC neurons' responses. Again, we looked at the positively tuned and the negatively tuned neuron populations separately. For the positively tuned neurons, their responses were significantly larger when the cue with higher value was rotated than when the cue with lower value was rotated (*Figure 4A*). The rotation effects were highly consistent at the level of individual neurons. Among 73 positively tuned neurons, 28 of them (36.4%) showed significant larger responses when the higher value cue was rotated than when the lower value cue was rotated. The mean responses to the higher and lower value cue conditions were significantly different (mean difference = 0.28, p=1.31e-4, two-tailed t-test) (*Figure 4C*). Only one neuron (1.3%) showed significant lower responses when the higher value cue was rotated.

A similar but opposite pattern was observed in the negatively tuned neurons. Consistent with their tuning, their responses were significantly lower when the cue with higher value was rotated than when the cue with lower value was rotated (*Figure 4B*). 12 of 37 (32.4%) showed a significant increase of responses when the lower value cue was rotated (*Figure 4D*). Only 2 (5.4%) neurons showed a response suppression. The mean responses to the higher and lower value cue conditions were significantly different (mean difference = −0.39, p=8.05e-6, two-tailed t-test) (*Figure 4D*).

The results from each individual monkey were consistent. The difference between the responses to the higher and the lower value cue rotation conditions was not significantly different between the two monkeys (two-tailed t-test, p=0.17 for positively tuned neurons and 0.17 for negatively tuned neurons, see also *Figure 4—figure supplement 1*).

Although the attentional modulation of each neuron's activity was highly consistent, the magnitude of attentional modulation was modest. We calculated the modulation index for each neuron. The mean modulation index for the positively and negatively tuned neurons were 0.05 and −0.07,

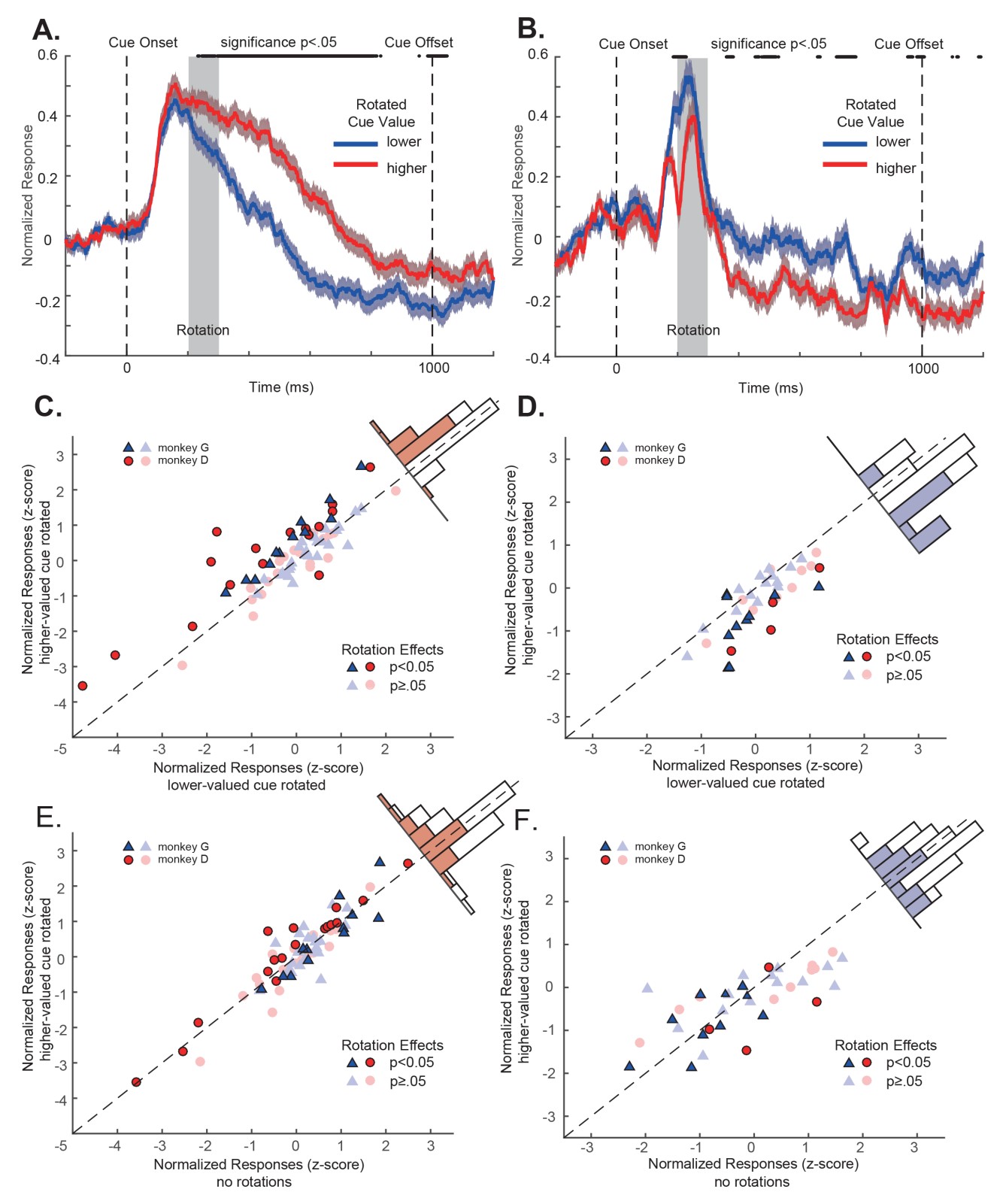

**Figure 4.** OFC responses to cue conditions with perturbations. (**A**) Population responses of the positively tuned OFC neurons to the double-cue conditions with visual perturbations. The blue curve includes the trials with the lower value cue rotated. The red curve includes the trials with the higher value cue rotated. The shading around each curve represents s.e.m. between trials. The grey box indicates the rotation period. The black dots on the top indicates the time points where the difference between two curves is significant (p<0.05 with multiple comparison corrections). (**B**)
*Figure 4 continued on next page*

*Figure 4 continued*

Population responses of the negatively tuned OFC neurons to the double-cue conditions with visual perturbations plotted in the same way as in panel A. (**C**) The comparison between each positively tuned neuron's responses when the higher and the lower value cues were rotated. Each data point represents a neuron. Bright data points are neurons that showed significant rotation effects, and dim data points are neurons that were not significantly affected by rotations. Blue triangles are neurons from monkey G, and Red circles are from monkey D. A histogram of the response differences between the two conditions is shown on the top right corner, in which filled squares indicate neurons with significant rotation effects. (**D**) Similar to C, but for the negatively tuned neurons. (**E**) The comparison between each positively tuned neuron's responses when the higher value cue was rotated and when there was no perturbation. Bright and dim data points are the same neurons with significant rotation effects as shown in panel (**C**). The color in the histogram also indicates the same significance as shown in panel C. (**F**) Similar to (**E**), but for the negatively tuned neurons.

DOI: https://doi.org/10.7554/eLife.31507.010

The following figure supplements are available for figure 4:

**Figure supplement 1.** Population responses to the double-cue conditions with visual perturbations of the OFC neurons from individual monkeys.

DOI: https://doi.org/10.7554/eLife.31507.011

**Figure supplement 2.** Attentional modulation indices of the value-encoding OFC neurons.

DOI: https://doi.org/10.7554/eLife.31507.012

**Figure supplement 3.** OFC responses to the cue conditions with perturbations.

DOI: https://doi.org/10.7554/eLife.31507.013

**Figure supplement 4.** Population responses to the double-cue conditions with visual perturbations of the OFC neurons from individual monkeys, with only the first 50 correct trials in each block were included.

DOI: https://doi.org/10.7554/eLife.31507.014

**Figure supplement 5.** OFC population responses to the double-cue conditions without visual perturbations (yellow) and with visual perturbations applied to the higher value cues (red).

DOI: https://doi.org/10.7554/eLife.31507.015

**Figure supplement 6.** The history of attentional modulation of OFC responses.

DOI: https://doi.org/10.7554/eLife.31507.016

**Figure supplement 7.** The distributions of the latencies and the durations of the attention modulation window in which the visual perturbation affected the neurons' responses significantly.

DOI: https://doi.org/10.7554/eLife.31507.017

**Figure supplement 8.** OFC neurons' responses to the double-cue conditions with perturbations applied to the higher value or the lower value cues.

DOI: https://doi.org/10.7554/eLife.31507.018

respectively. They were 0.13 and $-0.15$ for the neurons that showed significant attentional modulation (*Figure 4—figure supplement 2*).

The attentional modulation seemed to appear even before the onset of the visual perturbation. Other than a possible statistical fluke due to the limited amount of data, a possible explanation is that the monkeys might be able to predict the perturbation in some trials. Because the trials were arranged in blocks, the perturbations could in principle be predicted toward the end of blocks. Therefore, we analyzed the attentional modulation with the restriction of using only the first 50 trials from each block (*Figure 4—figure supplements 3,4*). The restriction largely eliminated the early attentional modulation. Our conclusion is, however, not undermined even if the monkeys were indeed using the block structure of the trials to prematurely orient their attention in some trials.

These results are consistent with the idea that the visual perturbation attracted monkeys' attention toward the perturbed cue, and the OFC neurons encoded its value. However, when there were no perturbations, we have suggested that the monkeys would attend to the higher value cue. If so, applying perturbations to the higher value cue should have minimal effects on the existing attention locations and the neurons' responses. Indeed, when we compared the neurons' responses when there were no perturbations and when the perturbations were applied to the higher value cue, we observed no significant change in the neurons' responses (*Figure 4EF*. p=0.82 and 0.21 for the positively and the negatively tuned neurons, respectively. See also *Figure 4—figure supplement 5*). Therefore, visual perturbations by themselves did not affect OFC neurons' responses. The attentional modulation that we observed when the perturbations were applied to the lower value cue was due to the attention shift toward the lower value cue.

## Attention shift and normalization model

To further demonstrate how attention shifts affected OFC neurons' responses when the perturbations were applied to the lower value cues, we looked at two specific cases of cue combinations.

The first case included conditions when the cue with the maximum reward, 8 drops of juice, was paired with another cue with rewards ranging from 0 to 8 drops of juice. When there were no visual perturbations, presumably the monkeys' attention was always on the cue with 8 drops of juice, and the OFC neurons' responses reflected that fact (red solid lines in *Figure 5AC*, and also in *Figure 5—figure supplement 1* for individual monkeys). When the visual perturbation was applied to the cue with less reward (red dashed lines in *Figure 5AC*, and also in *Figure 5—figure supplement 1* for individual monkeys), we saw a decrease of responses among the positively tuned neurons (p=6.10e-7, two-tailed paired t-test), and an increase of responses among the negatively tuned neurons (p=2.30e-5, paired two-tailed t-test), so that the population responses were driven toward the single cue conditions when only the lower value cue was presented (black solid lines in *Figure 5AC*, and also in *Figure 5—figure supplement 1* for individual monkeys). When the value difference between the two cues was large, such as in the 8 vs. 0 and 8 vs. 1 conditions, the shift was moderate. When the value difference between the two cues was small, such as in the 8 vs. 4 condition, the shift was closer to complete.

We can see a similar pattern in the second cue combination case. Here, we studied all conditions when the cue with the minimum reward, 0 drops of juice, was paired with another cue with reward ranging from 0 to 8 drops of juice. By default, the monkeys' attention was away from the cue with 0 drops of juice, and the OFC neurons' responses reflected the value of the other cue (p=0.0097 and 0.0077 for the positively and the negatively tuned neurons, respectively, one-way ANOVA with Bonferroni multiple comparison correction; blue solid lines in *Figure 5BD*, and also in *Figure 5—figure supplement 2* for individual monkeys). When the visual perturbation was applied to the 0-drops-of-juice cue, we saw a decrease of responses among the positively tuned neurons, and an increase of responses among the negatively tuned neurons (p=2.73e-5 and 3.67e-4 for the positively and the negatively tuned neurons, respectively, paired two-tailed t-test; blue dashed lines in *Figure 5BD*, and also in *Figure 5—figure supplement 2* for individual monkeys), so that the population responses were driven toward the condition when only 0-drops-of-juice cue was presented alone (black dashed lines in *Figure 5BD*, and also in *Figure 5—figure supplement 2* for individual monkeys). Again, the degree of attention shifts depended on the value difference between the two cues.

We modeled the attention shift based on the neurons' responses to single cues. We created a normalization model similar to those previously proposed to describe how neurons in the visual cortices respond to competing stimuli under different attention conditions (*Heeger, 1992*; *Ghose and Maunsell, 2008*; *Lee and Maunsell, 2009*; *Reynolds and Heeger, 2009*). The visual perturbation shifts the attention toward the perturbed cue. The key feature of our model is that the attention shift between the two cues is a function of the value difference between them. Attention should be shifted less when the value difference between the two options is large, because it is harder to drive the attention toward the lower value cue in this situation. We split the data of the neuronal responses in the double-cue conditions with visual perturbations into two halves and compared the model predictions based on parameters obtained from fitting half of the data (*Figure 5A–D*, green dashed lines) with the data from the other half (*Figure 5A–D*, red and blue dashed lines, see Methods). The model prediction matched the experimental data well. It explained 82.4% (s.e.m. = 1.46%) of the variance of the neuronal responses under the double-cue conditions with attention shifts caused by visual perturbations. The distributions of the best fitting parameters for each neuron are shown in *Figure 5—figure supplement 3*.

We compared the full mode against two other normalization models with fewer parameters. The key difference between these two reduced models and the full model is that the full model takes into account that the weight is a function of the value difference between the two cues. Again, we fit the three models by randomly dividing the trials into two halves. The fitting was based on one half of the dataset, and the obtained parameters were used to generate predictions to test against the other half of the data. We calculated the $R^2$ and AICc (Akaike information criterion with correction) for the predictions based on the test data set for each neuron. The full model performed significantly better than the reduced models, suggesting that the model that takes into account that the attention shift correlates with the value difference between the two cues explains the data best (*Figure 5—figure supplement 4*, and *Supplementary file 1*, Table 3).

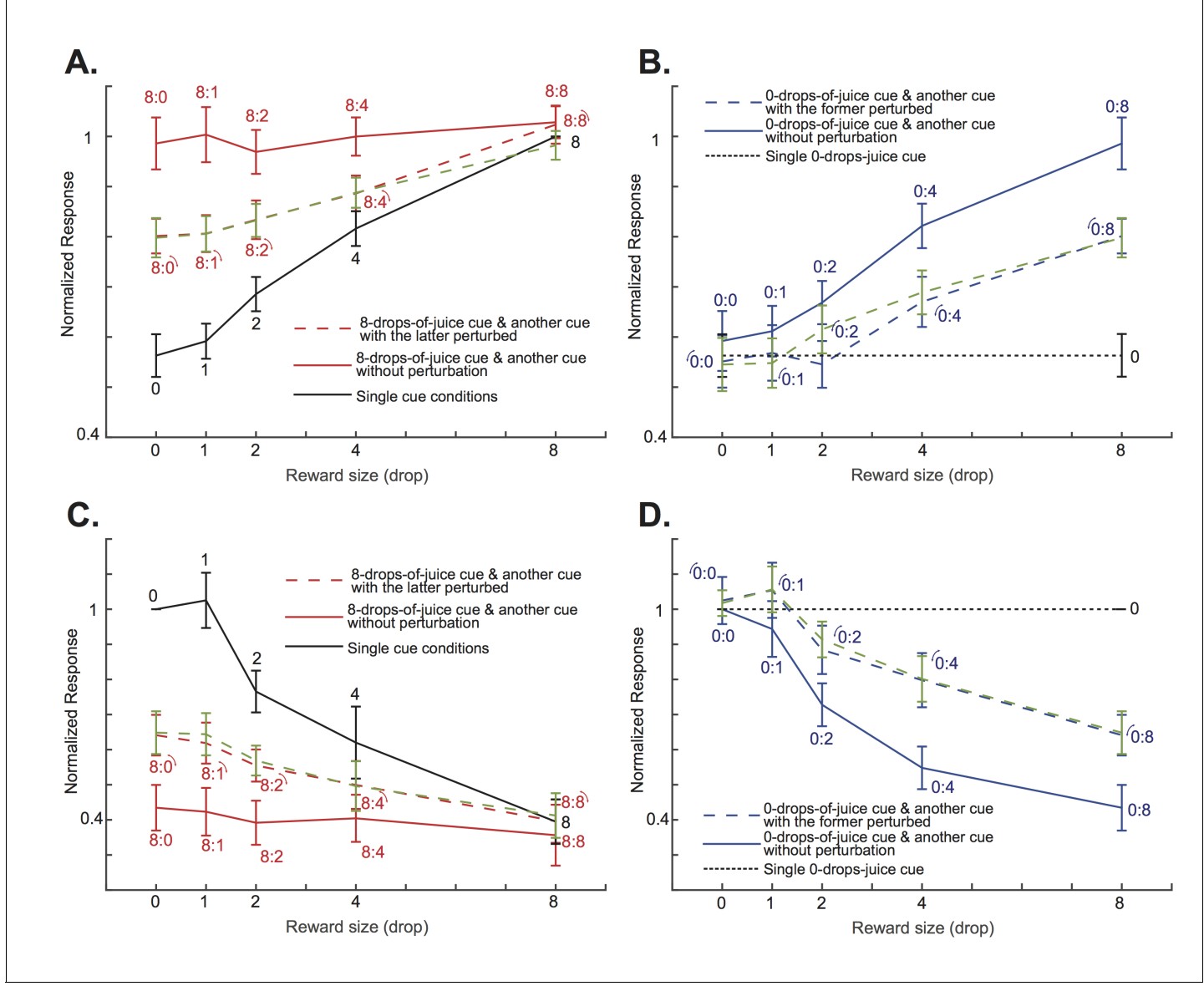

**Figure 5.** The attentional shift between the two cues depended on the difference between their associated value. (A) The solid red curve is the positively tuned OFC neurons' responses to an 8-drops-of-juice cue paired with another cue associated with 0, 1, 2, 4, or 8 drops of juice. The solid black curve indicates the responses to the lower value cue presented alone. The red dashed curve is the responses to the cue pairs with the visual perturbation applied to the lower value cue from the testing data set. The green dashed curve is the predicted responses of the testing dataset by the normalization model fitted with the training dataset. The numbers near each data point indicate the cue combinations that comprise each data point, with the little arcs indicating the perturbed cue. The error bars indicated s.e.m. between neurons. Notice that the responses were normalized to the responses under the neurons' preferred single-cue condition. Thus, there is no error bar for the 8-drops-of-cue condition. (B) The solid blue curve is the positively tuned OFC neurons' responses to a 0-drops-of-juice cue paired with another cue associated with 0, 1, 2, 4, or 8 drops of juice. The dotted black horizontal line indicates the responses to the single 0-drops-of-juice cue. The blue dashed line is the responses to the same cue pairs with the visual perturbation applied to the 0-drops-of-juice cue. The green dashed curve is the responses predicted by the normalization model. The numbers near each data point indicate the cue combinations that comprise each data point, with the little arcs indicating the perturbed cue. (C) and (D). The same analyses as panels (A) and (B), but for the negatively-tuned OFC neurons. All error bars indicate s.e.m. between neurons.

DOI: https://doi.org/10.7554/eLife.31507.019

The following figure supplements are available for figure 5:

**Figure supplement 1.** The modulation of the positively tuned neurons' responses due to the attentional shift between the two cues depended on the difference between their associated values.

DOI: https://doi.org/10.7554/eLife.31507.020

*Figure 5 continued*

**Figure supplement 2.** The modulation of the negatively tuned neurons' responses due to the attentional shift between the two cues depended on the difference between their associated values.

DOI: https://doi.org/10.7554/eLife.31507.021

**Figure supplement 3.** The distributions of the best fitting parameters in the full model of each neuron.

DOI: https://doi.org/10.7554/eLife.31507.022

**Figure supplement 4.** The comparison between the full model and model 2.

DOI: https://doi.org/10.7554/eLife.31507.023

## Discussion

### The passive-viewing task disentangles attention, value, and choice

Using a passive viewing paradigm was the key to our experimental design. It allowed us to tease apart reward, choice, eye movement and attention. First of all, in the double-cue condition, the reward was chosen randomly. It is clear that the OFC neurons did not encode reward expectation. Their responses to a pair of 8-drops-of-juice cues were statistically the same as to the pair of an 8-drops-of-juice cue and a cue associated with no reward (*Figure 5A*). The prospect of getting 8 drops of juice was very different under these two conditions. Second, when a visual perturbation was applied to the lower value cue, the expected reward remained the same. Thus, the only reasonable explanation for the observed change of OFC neural responses is the shift of attention toward the lower value cue. Third, because there was no active choice, we avoided the inevitable shift of attention toward the chosen item that accompanies a choice. The reward delivered to the monkeys was not contingent on the visual perturbation, and the monkeys never needed to make a choice. It is possible that monkeys were still covertly making choices and planning eye movements toward the perturbed cue, but additional analyses of eye positions do not support this scenario (*Figure 2—figure supplement 3*).

On the other hand, choice-related signals in the OFC that were found in several previous studies may be explained by attention (*Padoa-Schioppa and Assad, 2006*; *Padoa-Schioppa, 2013*; *Rudebeck et al., 2013a*; *Strait et al., 2014*). When choices are made, attention is most likely shifted toward the chosen option. Thus, the findings of the preponderance of OFC neurons encoding the value of chosen item over neurons encoding unchosen values during decision making (*Padoa-Schioppa and Assad, 2006*; *Padoa-Schioppa, 2013*) can be because the activity of these neurons reflected the attention shift toward the chosen item.

Although the observed attentional modulation of the neural responses in the OFC in this study was relatively modest, it represented a scenario in which the animals were not actively directing their attention toward a stimulus. They were passively viewing the stimuli without having to make any behavioral responses. The visual perturbation provided no information on the reward contingency. They could have just ignored the perturbation, and indeed the perturbation did not produce measurable behavior effects. In addition, the onset of the attentional modulation was after the peak visual responses of these neurons. Despite all these factors, we still observed a robust and consistent attentional modulation on a substantial proportion of value-sensitive OFC neurons (43 out of 110, or 39.1%). We speculate that the attentional modulation would be larger had the monkeys been engaged in a task that demands focused attention.

### A bottom-up attention mechanism

The attentional modulation we observed in the OFC was most likely due to a bottom-up instead of top-down attention mechanism. The monkeys did not have to actively direct its attention toward the visual perturbation. It is possible that the monkeys were doing that nevertheless. However, we found the strength of attentional modulation to be fairly consistent throughout the whole recording part of the experiment, which lasted 3 and 5 months for monkeys G and D, respectively (*Figure 4—figure supplement 6*). We would expect the modulation to have become smaller if it was due to a top-down control, because the monkeys would have been more likely to ignore the perturbation toward the end of the experiments.

In the double-cue condition without visual perturbations, the OFC neurons seem to encode the higher value cue from the very beginning. It is not known how and where in the brain it is

determined which cue is associated with higher value. We did not find evidence for an evolving decision in the OFC, calling into question whether the value-based decision making occurs in the OFC. However, the decision of where to pay attention in this experiment is simple, and it may be solved at a lower level of sensory or value information processing areas, such as the amygdala and the visual cortex (*Chelazzi et al., 1993*; *Peck et al., 2013*). After many months of training, the cues with high values probably gained greater salience than the cues with low values (*Anderson et al., 2011*). Thereby, a bottom-up attention mechanism may have picked out the cue with higher value and guided the OFC responses. The OFC may still play an important role when a value-based decision requires more deliberation and top-down control (*Beck et al., 2008*), as well as when values change and need to be retrieved at the time of choice (*Murray et al., 2015*).

## OFC and the visual system

As OFC neurons are not typically spatial selective, our findings can be most comparable to the previous studies in the visual system in which multiple stimuli were presented simultaneously within the same receptive field of a neuron. From this point of view, our findings are not unlike the findings in visual areas such as V4, MT and IT. When competing stimuli were presented inside the same receptive field of a neuron in the visual cortex, the neuron's responses to the attended stimulus were more similar to its responses when the attended stimulus was presented alone (*Chelazzi et al., 1993*; *Reynolds et al., 1999*; *Chelazzi et al., 2001*; *Ghose and Maunsell, 2008*; *Lee and Maunsell, 2009*, *2010*; *Ni et al., 2012*). However, when the attended stimulus was the non-preferred stimulus of an MT neuron, its responses still reflected the influence of the non-attended but preferred stimulus (*Ni et al., 2012*). In contrast, our results showed that even the negatively tuned OFC neurons' responses were completely dominated by higher value cues, which were their non-preferred stimuli. Thus, the normalization models proposed for the visual system in the previous studies of the visual cortex do not explain the behavior of OFC neurons.

In our normalization model, we modeled the OFC neural responses to multiple items as a weighted average of its responses to each individual item. Although the model may appear similar to the previously described normalization models of attention effects in the visual system (*Reynolds et al., 1999*; *Ghose and Maunsell, 2008*; *Lee and Maunsell, 2009*; *Reynolds and Heeger, 2009*; *Lee and Maunsell, 2010*; *Ni et al., 2012*), we interpret our normalization model differently. We believe that the weighting is done across the time rather than across the space. The weights reflect how often attention is directed to an item. At any given time, the attention works in a winner-take-all fashion. Admittedly, we only observed such winner-take-all attention effects in the cue conditions without visual perturbations where the attention was on the higher value stimulus. However, the fact that most OFC neurons, regardless of its value preference, showed such consistent winner-take-all modulation effects strongly suggests that the winner-take-all mechanism works under other stimulus conditions as well.

The homogenous attentional modulation within the OFC population also leads us to believe that the attentional modulation that we observed in the OFC is inherited from the visual cortex. It has been reported that the OFC neurons had lower noise correlations than neurons in the visual cortex (*Padoa-Schioppa, 2013*; *Conen and Padoa-Schioppa, 2015*). Furthermore, It has been found that attention decreases noise correlation in the visual cortex (*Cohen and Maunsell, 2009*). As we showed that the OFC neurons only encoded the value of attended stimuli, the lower noise correlations between the OFC neurons may be expected if they receive inputs from the population of visual neurons that represent attended stimuli.

## Serial processing

Our results are consistent with the hypothesis that the OFC encodes the value information of one item at a time even when multiple items are presented. Several previous studies have provided additional evidence in favor of the hypothesis. *McGinty et al. (2016)* found that the value coding in the OFC was highest when the animals were fixating at locations near the cue. Although there was only one cue presented to the animal in the study, their results indicated that eye fixation and overt attention could modulate the value coding in a similar manner as in the case of covert attention. *Rich and Wallis (2016)* showed that a single OFC population's activity alternated between states associated with the value of available options. It was not known if the alternation can be explained

by attention shifts. Yet, the study provided important evidence that OFC evaluated the value of each option sequentially. Finally, it is reported that human subjects solved a multi-cue probabilistic classification task by integrating different numbers of simultaneously presented cues under varying time pressure, suggesting a sequential processing was in play (*Oh et al., 2016*). Taken together, the current evidence supports the hypothesis that the value information is processed in the brain in a sequential manner that is guided by attention and reflected by the OFC activity.

The findings of offer-value neurons in the OFC (*Padoa-Schioppa and Assad, 2006*; *Padoa-Schioppa, 2011*), which encode values of both reward options that are tied to specific juice types, may appear to argue against our hypothesis and suggest that the values of both options are simultaneously represented in the OFC. Yet these findings are potentially compatible with the current study. The current study did not use different juice types as rewards. Future studies are required to find out how attention modulates OFC neurons' responses when the animals attend to stimuli leading to different reward types.

### Attention and value-based decision making

If the OFC underlies value-based decision making as previously suggested (*Padoa-Schioppa, 2011*), our results favor the idea that value-based decision making is a sequential process in which the value of each option is evaluated in the OFC one at a time, guided by attention. It was shown that value-based decision making may be driven by the value difference between attended and unattended stimuli, which was found to be represented by the neural activity in human vmPFC and ventral striatum (*Krajbich et al., 2010*; *Lim et al., 2011*). However, we demonstrated that the OFC activity in monkeys represented only the higher value between the alternatives instead of the value difference between attended and unattended stimuli. It remains a question whether the discrepancy between our study and the previous human fMRI studies is due to species difference or the difference between neural signals recorded with fMRI and with electrophysiology.

Rangel and his colleagues (*Armel et al., 2008*; *Krajbich et al., 2010*) also proposed that fixation or attention may bias value-based decision making by assigning a larger weight to the attended option's value during decision making and the OFC was underlying this fixation bias. Were it the case, we should observe that the positively tuned OFC neurons to have higher responses toward the attended item and the negatively tuned neurons to have lower responses. Our results did not support this scenario. Recently, it was reported that patients with ventromedial frontal lesions showed deficits in directing attention based on cue-reward associations but exhibited fixation biases during value-based decision similar to health controls (*Vaidya and Fellows, 2015a*,*Vaidya and Fellows, 2015b*). Thus, it is possible that downstream structures in the brain that integrate value signals from the OFC reflect this attention bias.

### Summary

Based on these results, we may speculate how the OFC supports value-based decision making when multiple items are presented simultaneously. During decision making, the brain evaluates the value of each item sequentially, and the visual features of each item play an important role in guiding the attention to and away from each item. The OFC activity encodes the value of the attended item during this process. Other downstream brain areas may extract the information encoded in the OFC, compare the value between the options, and carry out the decision making. Many interesting details in this speculation are still missing but can be addressed in future studies.

## Materials and methods

### Subjects and materials

We trained two naive male rhesus monkeys (*Macaca mulatta*) in the study. They weighed 6.4 and 7.4 kg at the beginning of the experiment. All experimental procedures were approved by the Animal Care Committee of Shanghai Institutes for Biological Sciences, Chinese Academy of Sciences (Shanghai, China).

During experiments, the monkeys were seated in a primate chair facing a 23.6-inch video monitor. Rewards consisted of 1 to 8 drops of juice per trial (0.08 ~ 0.5 ml), with drop size controlled by a

computer-controlled solenoid. Eye positions and pupil size measurements were monitored with an infrared oculometer system at a sampling rate of 500 Hz (EyeLink 1000).

## Behavioral task

Two monkeys G and D were trained to perform a passive viewing task (*Figure 1A*). Each trial began with the appearance of a fixation point in the center of the screen. After the monkeys gazed at the fixation point for 2000 ms, one or two simple geometric shapes of 2.5° in size were presented at 7° away from the fixation point for monkey G and 10° away for monkey D at horizontal positions on a computer screen for 1000 ms. When the cues were extinguished, there was a delay period of 1500 ms, at the end of which a reward was delivered if the monkeys held their fixation successfully. The fixation window was 2° in size for monkey G and 3° for monkey D. If the monkeys broke their fixation, a penalty timeout of 4000 ms was added.

The visual cues informed the monkeys of the number of drops of juice that would be delivered. The cue set contained five cues, each associated with 0, 1, 2, 4, and 8 drops of juice, respectively. When there was only one cue presented on the screen, its associated reward would be delivered. When two cues were simultaneously presented, one of the cues was randomly selected and its associated reward delivered. Two different cue sets were used for the two monkeys, respectively (*Figure 1B*).

In some trials with two cues, a visual perturbation was added to one of the cues. The perturbation was a quick back-and-forth rotation of 90° for most of the cues and 45° for the square and the star to make rotations easier to see. Its onset was 200 ms after the cue onset and lasted for 100 ms. The perturbation was independent of the assignment of reward and thus did not provide the monkeys any information on the upcoming reward.

All cue conditions were interleaved in blocks. There were 85 conditions in each block: 10 single-cue conditions with each of the 5 cues appearing on either the left or the right, 25 double-cue conditions (each of the 5 cues could appear on the left or right side), and 50 double-cue conditions with visual perturbations applied to either the left or the right cue. Each block contained one trial from each condition. The conditions were randomly interleaved, and the monkeys had to complete all conditions in a block before they started a new one. During recording experiments, the monkeys on average completed 9.4 blocks daily.

## Surgery

At the beginning of the training, both monkeys received a chronic implant of a titanium headpost with standard procedures. After recovery, the monkeys received training for the main task until their performance was satisfactory. Then we performed a second surgery to implant an acrylic recording chamber over the prefrontal region. A craniotomy was made inside the chamber. All surgeries were performed under aseptic conditions. The monkeys were sedated with ketamine hydrochloride (5–15 mg/kg, i.m.), and anesthesia was then induced and maintained with isoflurane gas (1.5–2%, to effect). Body temperature, heart rate, blood pressure, and expired $CO_2$ were monitored throughout all surgical procedures.

## MRI

Before and after the recording chamber was implanted, we acquired structural Magnetic Resonance Imaging (MRI) scans to identify and then verify implant locations. Scans were carried out on a Siemens 3T scanner. Monkeys were sedated with ketamine hydrochloride (5–15 mg/kg, i.m.), and anesthesia was then induced and maintained with isoflurane gas (1.5–2%, to effect).

## Electrophysiology

We recorded single unit activity with vertically movable electrodes (Alpha Omega and FHC, 0.5–1.5 MΩ at 1 KHz) using conventional techniques. Briefly, microelectrodes were driven by a multi-channel micromanipulator (Alpha Omega EPS) attached to the recording chamber. At most four electrodes were inserted at the same time. Recording locations were on the ventral surface of the frontal lobe between the lateral and medial orbital sulci, roughly corresponding to Walker's areas 11 and 13 (*Walker, 1940*). Spike waveforms from putative single neurons were isolated online and recorded

with an Alpha Omega SnR system. Offline sorting was done with NeuroExplorer. Other than the quality of isolation, there were no selection criteria for neurons.

## Pupil dilation analysis

The pupil responses were recorded during all recording sessions. For consistency, the analyses were based on the sessions in which value-encoding neurons were identified and analyzed (33 and 38 sessions from monkey D and G, respectively). We calculated the average pupil size from 400 to 900 ms after the monkeys acquiring fixation as the baseline. The baseline was then subtracted from the pupil responses in the rest of the analyses. To quantify how reward expectation affected pupil dilation, we calculated the average pupil size in the 1 s period after the offset of the cue. During this period, the pupil dilation showed a consistent pattern between the two monkeys.

We quantified how reward value affected pupil dilation in the single-cue conditions with the following linear regression:

$$PS = b_0 + b_1 \bullet V, \tag{1}$$

where PS is the pupil size measured in the aforementioned window, V is the value (number of drops of juice) associated with the reward cue.

A two-way ANOVA was used to test the pupil dilation difference between the single-cue conditions and the double-cue conditions grouped by the higher value, with one factor being the reward size and the other the number of cues. Post-hoc analyses were done with Tukey tests.

## Electrophysiology analysis

The electrophysiology data were based on 84 and 67 recording sessions from monkeys G and D, respectively. On average, 5.6 neurons were recorded from each session, and 297 trials were recorded from each neuron.

### Value selectivity

We first categorized a neuron as visually responsive if its mean firing rate between the fixation onset to the cue onset was significantly different from that between the cue onset to the cue offset based on a t-test. To determine whether a neuron was selective to value, we calculated its average responses between 150 and 550 ms after the cue onset in the single-cue conditions, as most of the neurons had a visual latency larger than 150 ms and their responses were transient. We then used a one-way ANOVA to determine the significance at p<0.05. Further analyses were only performed on the value-selective neurons.

We further divided the value-selective neurons into the positively and the negatively tuned groups with a linear regression:

$$FR = b_0 + b_1 \bullet V, \tag{2}$$

where FR is the neuron's cue responses during the single-cue conditions and V indicates the cue's associated value (V = 0, 1, 2, 4, or 8). The significance was determined at p<0.05. We assigned the neurons to the positively and negatively tuned groups according to the sign of $b_1$.

### Population responses

In *Figure 3C–E*, each neuron's responses were normalized to its response to its preferred single cue condition (8-drops-of-juice for the positively tuned group and 0-drops-of-juice for the negatively tuned group) after the baseline was subtracted. The mean response in the 200 ms time window before the cue onset was used as the baseline. The population responses were calculated as the average of the normalized responses of each neuron. The particular choice of normalization method did not change the conclusions.

### Visual perturbation analyses

In the PSTHs in *Figure 4AB*, each neuron's PSTH was calculated with a sliding window of 50 ms and averaged across all completed trials. We used a 200 ms window before the cue onset to calculate the baseline and subtracted the neuron's response baseline before normalizing the responses to

each neuron's peak response to the preferred condition. The population responses were the average of the normalized responses of each neuron. The significance of the difference between the two rotation conditions in *Figure 4AB* was determined by a two-tailed t-test performed at each time point. To account for multiple comparisons, we used the Benjamini-Hochberg procedure to control the false discovery rate to be under 0.05 (*Hochberg and Benjamini, 1990*). The analyses for individual monkeys are plotted in *Figure 4—figure supplement 1*.

We determined whether the visual perturbation affected a neuron's responses in *Figure 4CD* as follows. For each unit, all completed trials with visual perturbations were divided into two groups. One included the conditions when the cue with the higher value was rotated, and the other included the conditions when the lower value cue was rotated. We defined a neuron's responses as significantly modulated by rotations if there was a time window no less than 200 ms during the period between the onset of the rotation and the offset of the cue (200–1000 ms after the cue onset) in which the mean firing rates between the two groups were significantly different when tested with a two-tailed t-test. And this time window was defined as its modulation window (*Figure 4—figure supplement 7*). For the units that were significantly modulated by rotation, the Z scores of their firing rates during the modulation window were calculated and plotted in *Figure 4C–F*. For the units without significant rotation effects, we plotted the Z scores of their firing rates between the onset of the rotation to the offset of the cue. For *Figure 4EF*, the responses under conditions without rotations were calculated using the same time window as the corresponding rotation conditions. The analyses plotted in *Figure 4CD* are also plotted in *Figure 4—figure supplement 8* with a uniform time window for all neurons, which is from 450 to 750 ms after the cue onset (see also *Supplementary file 1*, Table 1 for detailed statistics). The specific choice of statistical analyses does not affect the conclusion.

Because the trials were blocked, the monkeys might be able to predict upcoming rotation conditions toward the end of a block. To minimize this potential confounder, we repeated the analyses in *Figure 4AB*, but using only the first 50 correct trials from each block in *Figure 4—figure supplement 3*. To make sure there are same numbers of trials of each cue in a particular cue combination being perturbed and non-perturbed, we randomly removed trials from the perturbation condition that has more trials to make the values of the perturbed and non-perturbed cues even. Similarly, the analyses for individual monkeys were repeated in *Figure 4—figure supplement 4*.

We defined each neuron's attentional modulation index as:

$$\mathrm{MI} = \frac{(FR_h - FR_l)}{(FR_h + FR_l)}, \tag{3}$$

where MI is the modulation index, $FR_h$ and $FR_l$ are the neuron's responses when the higher and lower value cues were rotated, respectively.

## Shift of attention analyses

The population responses of positively tuned neurons in *Figure 5AB* were calculated as the mean firing rate between the onset of the perturbation and the offset of the cue (200–1000 ms after the cue onset) normalized to each neuron's responses to its preferred single-cue (8 drops-of-juice) condition. Similarly, the population responses of negatively tuned neurons in *Figure 5CD* were mean firing rates between the onset of the perturbation and the offset of the cue normalized to each neuron's responses to its preferred single-cue (0 drops-of-juice) condition. The analyses in *Figure 5—figure supplements 1* and *2* were done similarly.

## Normalization model

For each neuron, we modeled their responses under double-cue conditions with their responses under single-cue conditions. We calculated each neuron's mean firing rates under these conditions in a time window between 200 and 1000 ms after cue onset, which were then normalized to the neuron's response to its preferred single-cue condition (8-drops-of-juice cue for positively tuned neurons, and 0-drops-of-juice cue for negatively tuned neurons).

We modeled each neuron's firing rate $R$ under double-cue conditions as follows:

$$R = (1-a) \cdot R_h + a \cdot R_l + b \tag{4}$$

where $R_h$ and $R_l$ are the neuron's responses (as plotted in *Figure 5*) under single-cue conditions when the higher value cue and the lower value cue is presented alone, respectively; $0 \leq a \leq 1$ is a parameter that indicates how attention is distributed, and $b$ is a constant.

When there was a visual perturbation applied to the lower value cue, we modeled the shift of attention toward the lower value cue as a sigmoid function:

$$a = d/(1 + \exp(c \bullet |\mathrm{R}_h - \mathrm{R}_l|)), \tag{5}$$

where $0 \leq c \leq 100$ and $0 \leq d \leq 2$ are free parameters that keep $a$ in between 0 and 1 and describe how attention is distributed between the two cues as a function of their value difference. Thus, there are totally three free parameters in our model: $b$, $c$, and $d$. We fit the model to the neurons that were found to be congruently and significantly modulated by visual perturbations (n = 28 for the positively tuned neurons, n = 12 for the negatively tuned neurons). For each neuron, we randomly divided the trials into two halves. The fitting was based on one half of the dataset with the least-squares method and the obtained three parameters were used to generate predictions to test against the other half of the data (green dashed curves in *Figure 5*). The reported explained variance reflected the average $R^2$ across all neurons.

In addition, we tested the full model against the following two reduced models.

Model 1. In this model, we modeled the neurons' responses to the double-cue conditions with the visual perturbation applied to the lower value cue as a weighted average of their responses to each cue. In addition, the weight $a$ is the same for all neurons and is independent of the value difference between the two cues:

$$\mathrm{R} = (1-a) * \mathrm{R}_h + a * \mathrm{R}_l + b, \tag{6}$$

The best fits for $a$ are 0.43 and 0.46 for the positively and negatively tuned neurons, respectively.

Model 2. Similar to Model 1, the neurons' responses to double cues under the rotation condition are modeled as a weighted average of their responses to each cue that is independent of the value difference between the two cues. However, we allow the weight $a$ to vary between neurons. The rest of the Model two is the same as in Model 1.

## Acknowledgements

This work was supported by the CAS Hundreds of Talents Program, and by Science and Technology Commission of Shanghai Municipality (15JC1400104). We thank Elisabeth Murray and Peter Rudebeck for comments on the manuscript, and Cheng Chen, Yang Chen, Yuanfeng Zhang, Zhongqiao Lin, Zhewei Zhang, and Wei Kong for their help in all phases of the study.

## Additional information

### Funding

| Funder | Grant reference number | Author |
| --- | --- | --- |
| Chinese Academy of Sciences | Hundreds of Talents Program | Tianming Yang |
| Science and Technology Commission of Shanghai Municipality | 15JC1400104 | Tianming Yang |

The funders had no role in study design, data collection and interpretation, or the decision to submit the work for publication.

### Author contributions

Yang Xie, Software, Formal analysis, Validation, Investigation, Visualization, Writing—original draft, Writing—review and editing; Chechang Nie, Formal analysis, Validation, Investigation, Writing—review and editing; Tianming Yang, Conceptualization, Resources, Data curation, Software, Formal

analysis, Supervision, Funding acquisition, Validation, Investigation, Visualization, Methodology, Writing—original draft, Project administration, Writing—review and editing

### Author ORCIDs
Tianming Yang (iD) http://orcid.org/0000-0001-6976-9246

### Ethics

Animal experimentation: All experimental procedures were approved by the Animal Care Committee of Shanghai Institutes for Biological Sciences, Chinese Academy of Sciences (ER-SIBS-221501P). All surgeries were performed under aseptic conditions. Monkeys were sedated with ketamine hydrochloride (5-15 mg/kg, i.m.), and anesthesia was then induced and maintained with isoflurane gas (1.5-2%, to effect). Every effort was made to minimize suffering.

### Decision letter and Author response
Decision letter https://doi.org/10.7554/eLife.31507.027
Author response https://doi.org/10.7554/eLife.31507.028

## Additional files

### Supplementary files
• Supplementary files 1. includes Supplementary Tables 1–3.
DOI: https://doi.org/10.7554/eLife.31507.024

• Transparent reporting form
DOI: https://doi.org/10.7554/eLife.31507.025

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
