## [Decision Letter]

Thank you for submitting your article "Covert Shift of Attention Modulates the Value Encoding in the Orbitofrontal Cortex" for consideration by *eLife*. Your article has been reviewed by three peer reviewers, and the evaluation has been overseen by a Guest Reviewing Editor and Michael Frank as the Senior Editor. The following individual involved in review of your submission has agreed to reveal his identity: Camillo Padoa-Schioppa (Reviewer #3).

The reviewers have discussed the reviews with one another and the Reviewing Editor has drafted this decision to help you prepare a revised submission.

Summary:

This study is a novel attempt to bring together two broad threads of cognitive and systems neuroscience – value representation and attentional mechanisms. It also adds to the literature addressing serial mechanisms of evidence evaluation and choice, a topic of growing interest (e.g. Shohamy and Shadlen, 2015). An additional strength of this study is that the experiment attempts to explicitly control the two major variables that are often entangled: reward expectation and attention; the use of a non-choice paradigm also removes another potentially confounding variable. The overall statistical approach is appropriate, and the graphical and written presentation is clear. However, there were a number of issues that reviewers would like to see addressed before deciding on the suitability of the manuscript for publication.

Main comments:

1) All reviewers were concerned about the weak neural effects. It is particularly important that rigorous statistical methods are employed to determine whether the effects are real. For example, all reviewers were in agreement that one-tailed tests could not be justified. In addition, there was concern that the effects in Figure 4 are apparent before the stimulus was rotated. This might potentially be a smoothing artifact, but the smoothing window is not reported, making it difficult to determine whether this is indeed the case.

2) Reviewers were not convinced by the modeling component of the manuscript. One possibility is that the neural effects arise as a consequence of dividing attention to the two stimuli when the lower value is perturbed. Thinking of the effect as resulting from divided attention, the firing rate in the rotation trials would simply equal the (possibly weighted) average between those associated with the two cues. This account would not need any free parameter (or at most one capturing the weights) and it is very consistent with the data. This would be a simpler model than the selective attention model with four free parameters. The authors should do a formal model comparison with their chosen model against these potentially simpler models, to show that the more complex model is a better predictor of neural activity.

3) Several points made in the Discussion were unconvincing or problematic.

a) "Although the observed attentional modulation of the neural responses in the OFC in this study was relatively modest, it represented a worst-case scenario." One could make the opposite claim: If the behavioral task had required using values computed or represented in the OFC, monkeys would have considered both options more evenly. In other words, contrary to the authors speculation, the attentional modulation might have been smaller had the monkeys been engaged in a choice task.

b) The authors discuss the neuronal effects described here as "explained by attention". However, the effects recorded here differ in fundamental ways from the attentional effects often described in sensory regions. Normally, the effect of attention on the activity of a neuron is understood as a gain modulation on a sensory response. For example, neurons in visual areas have receptive fields; if the animal pays attention to a particular location, the visual responses of neurons whose RF coincides with that location are enhanced. Concurrently, the visual responses of neurons with different RFs are somewhat reduced. In contrast, OFC cells are not spatially selective – that is, they don't have a receptive field at all. (This is fundamental difference between OFC and V4, MT, etc. Hence, the first paragraph of the subsection “OFC and the visual system” is incorrect.) Thus the neuronal effects found here may have to do more with mental focus than with attention as defined in the context of sensory systems, even though at the behavioral level the effects are driven by a shift of attention.

c) The authors re-interpret previous findings of neurons coding the chosen value as neurons coding the value the animal was focusing on. This may be an acceptable interpretation. Then they write "Our results suggest that the OFC encodes the value information one item at a time even when multiple items are presented" and "our results suggest that value-based decision making is a sequential process in which the value of each option is evaluated one at a time, guided by attention". These statements are completely speculative and they seem at odds with previous results. Along with neurons coding the chosen value, several studies found neurons coding the value of individual offers, or offer values. Furthermore, these cells have low noise correlations and low choice probabilities (Conen and Padoa-Schioppa, 2015). If trial-by-trial fluctuations in the activity of these neurons and behavioral choice variability were both driven by attention, one would expect much higher measures for both noise correlation and for choice probability. Thus it might be the case that only some of the value coding neurons in OFC are subject to the effects described here.

[Editors' note: further revisions were requested prior to acceptance, as described below.]

Thank you for resubmitting your work entitled "Covert Shift of Attention Modulates the Value Encoding in the Orbitofrontal Cortex" for further consideration at *eLife*. Your revised article has been favorably evaluated by Michael Frank (Senior Editor), a Guest Reviewing Editor, and three reviewers.

The manuscript has been improved but there are some remaining issues that need to be addressed before acceptance, as outlined below. We do not ordinarily invite multiple rounds of revision so please consider this the last opportunity you will have to respond to the concerns expressed below.

In general, there is agreement that the manuscript's findings are potentially important, but there are serious concerns about the statistical analysis that underlies the main findings of the paper. Since the size of the effects is on the smaller side, it is critical that rigorous statistical tests are used (two-tailed and corrected for multiple comparisons) and that the potential artifact can be satisfactorily explained, otherwise we will not be able to accept the paper.

*Reviewer #1:*

My thanks to the authors for addressing many of the concerns. While this manuscript is improved, there are still very serious issues that have not been resolved.

The most significant issues concern Figure 4 which shows how neural activity changes around the time of cue rotation. In brief: there is a lack of explanation for early onset of the effects; there is still a lack of statistical rigor in the significance testing; and there is a glaring data discrepancy between the primary and supplemental figures. Given these issues, I don't have confidence in these data, which are the main results of the paper. All three of these deficiencies – the first being the most critical – must be addressed. In more detail:

1) The original version of Figure 4 showed spurious neural activity changes before rotation. While the pre-rotation effects have largely disappeared, there are still neural effects during rotation, and these are still too early to be caused by the rotation. The authors must offer some plausible explanation for this. While they claim it could be due to noise, this is unlikely, as the effect is evident in both monkeys (Figure 4—figure supplement 1), for both positive and negative coding neurons.

To elaborate: In Figure 4, the latency to a significant effect is 30ms, which the authors attribute to a "bottom-up" process. However, the initial response to cue onset (presumably also a bottom-up process) occurs at a much slower latency of approximately 70-80ms. The situation is even worse in Figure 4, where the effects are significant at the onset of cue rotation at t=200ms. This first significant bin spans from t=175 to t=225ms, meaning that it includes data from the first 25ms after rotation begins. This is far too little time for rotation-induced attentional effects to appear, given that minimum visual response latencies within the lateral geniculate nucleus are approximately 20ms (Schroeder CE et al. Brain Res. 1989 Jan 16;477(1-2):183-95.).

If this were my data, I would strongly suspect an artifact of some kind. For example, if the trial conditions were insufficiently randomized, it is possible that the monkeys could anticipate which stimulus was going to be rotated on some trials, so that they could shift attention before the rotation begins.

2) In Figure 4 a one-tailed test is not justified. If this were a follow-on study looking to confirm a previously documented effect, then the authors could claim a priori justification for using a one-tailed test. But this study is an investigation of a novel phenomenon, and so the threshold for rejecting the null hypothesis must be suitably strong. While Figure 4—figure supplement 5 uses a two-tailed test, it does not, according to the legend, correct for multiple comparisons. For this analysis to be credible, it must be performed with two-tailed tests, with some form of p-value or FDR correction.

3) In Figure 4 there is a discrepancy between the main and supplemental figures. The same data are supposedly plotted in Figure 4 and Figure 4—figure supplement 5 the only difference being the use of two-tailed tests to draw the significance indicators at the top of the plot. However, the means are clearly different between the two figures. For example, in Figure 4, the red and blue lines overlap at ~850ms, but in Figure 4—figure supplement 5 they do not. These are clearly different data.

*Reviewer #2:*

I have reviewed the revised manuscript. For the most part, I believe the authors have addressed concerns raised in the first review. My major outstanding concern is that the 1-tailed tests in Figure 4 are still not appropriate. In panels C-F the authors report neurons with significant effects in directions opposite of their a-priori hypothesis, therefore they are in fact testing both sides of the distribution, but using a 1-sided significance threshold. The authors have the relevant 2-tailed stats and figures in the supplement, and should use these in the main paper.

I think the addition of alternative models has strengthened their results. However, in the table of model parameters, it says that models 1 and 2 have 1 and 2 free parameters respectively. I'm not sure this is correct. For example, model 1 fits a and b, and in the calculation of AICs, the intercept should be included as a parameter, so there should be 2. This shouldn't change the outcome, but stands out as a potential error.

On this note, it seems odd to include the constant, b, in these models at all, when FRs of each neuron are being directly fit from their own FRs. In other words, the underlying hypothesis is that a neuron's FR in the double-cue condition is some mixture of its own FRs in the two single-cue conditions – not that mixture with a constant offset. So why include the constant?

*Reviewer #3:*

The manuscript was significantly improved. However, a couple of points raised in the first review require additional clarification. Numbers refer to those in the first review.

3a) Apropos the idea that the attentional modulation observed in this study represented a "worst-case scenario": In the initial review, it was noticed that one could make the opposite speculation. In a choice setting, which requires using values computed in OFC, monkeys would have considered both options more evenly. Hence, contrary to the authors assertion, had the monkeys been engaged in a choice task, the attentional modulation might have been smaller, not larger. The authors did not take this issue at heart, but they should. Let me note that this is more than a simple discussion point: As the authors make clear in their response to point (1), the worst-case scenario argument is part of why they think the small effects described in this paper are real. Again, I don't see any a-priori reason to think that in a choice setting the attentional modulation would be stronger. If this claim is so important, and if it relies on unpublished data, the data should be described in this paper and we should be given the opportunity to review the evidence. If this claim is not so important, the authors should drop it here and make it elsewhere, when they present the appropriate evidence. In the latter case, they might want to provide a different response to point (1).

3c) If I understand correctly, the authors speculate that offer value cells provide an indirect input to the decision circuit, which operates with working memory mechanisms. However, the fact that offer value cells exist (i.e., the fact that in choice settings some neurons are associated to individual offers), and the fact that noise correlations between them are small implies that these cells are not modulated by attention in the sense described in this manuscript. Do the authors agree on this point? The was the sense of my previous comment, not whether the contribution of offer value cells to the decision is direct or indirect.

[Editors' note: further revisions were requested prior to acceptance, as described below.]

Thank you for resubmitting your work entitled "Covert Shift of Attention Modulates the Value Encoding in the Orbitofrontal Cortex" for further consideration at *eLife*. Your revised article has been favorably evaluated by Michael Frank (Senior Editor), a Reviewing Editor, and three reviewers.

The manuscript has been improved but there are some remaining issues that need to be addressed before acceptance, as outlined below. We do not ordinarily invite multiple rounds of revision so please consider this the last opportunity you will have to respond to the concerns expressed below.

In general, there is agreement that the manuscript's findings are potentially important, but there are serious concerns about the statistical analysis that underlies the main findings of the paper. Since the size of the effects is on the smaller side, it is critical that rigorous statistical tests are used (two-tailed and corrected for multiple comparisons) and that the potential artifact can be satisfactorily explained, otherwise we will not be able to accept the paper.

*Reviewer #1:*

My thanks to the authors for addressing many of the concerns. While this manuscript is improved, there are still very serious issues that have not been resolved.

The most significant issues concern Figure 4 which shows how neural activity changes around the time of cue rotation. In brief: there is a lack of explanation for early onset of the effects; there is still a lack of statistical rigor in the significance testing; and there is a glaring data discrepancy between the primary and supplemental figures. Given these issues, I don't have confidence in these data, which are the main results of the paper. All three of these deficiencies – the first being the most critical – must be addressed. In more detail:

1) The original version of Figure 4 showed spurious neural activity changes before rotation. While the pre-rotation effects have largely disappeared, there are still neural effects during rotation, and these are still too early to be caused by the rotation. The authors must offer some plausible explanation for this. While they claim it could be due to noise, this is unlikely, as the effect is evident in both monkeys (Figure 4—figure supplement 1), for both positive and negative coding neurons.

To elaborate: In Figure 4, the latency to a significant effect is 30ms, which the authors attribute to a "bottom-up" process. However, the initial response to cue onset (presumably also a bottom-up process) occurs at a much slower latency of approximately 70-80ms. The situation is even worse in Figure 4, where the effects are significant at the onset of cue rotation at t=200ms. This first significant bin spans from t=175 to t=225ms, meaning that it includes data from the first 25ms after rotation begins. This is far too little time for rotation-induced attentional effects to appear, given that minimum visual response latencies within the lateral geniculate nucleus are approximately 20ms (Schroeder CE et al. Brain Res. 1989 Jan 16;477(1-2):183-95.).

If this were my data, I would strongly suspect an artifact of some kind. For example, if the trial conditions were insufficiently randomized, it is possible that the monkeys could anticipate which stimulus was going to be rotated on some trials, so that they could shift attention before the rotation begins.

2) In Figure 4 a one-tailed test is not justified. If this were a follow-on study looking to confirm a previously documented effect, then the authors could claim a priori justification for using a one-tailed test. But this study is an investigation of a novel phenomenon, and so the threshold for rejecting the null hypothesis must be suitably strong. While Figure 4—figure supplement 5 uses a two-tailed test, it does not, according to the legend, correct for multiple comparisons. For this analysis to be credible, it must be performed with two-tailed tests, with some form of p-value or FDR correction.

3) In Figure 4 there is a discrepancy between the main and supplemental figures. The same data are supposedly plotted in Figure 4 and Figure 4—figure supplement 5, the only difference being the use of two-tailed tests to draw the significance indicators at the top of the plot. However, the means are clearly different between the two figures. For example, in Figure 4, the red and blue lines overlap at ~850ms, but in Figure 4—figure supplement 5 they do not. These are clearly different data.

*Reviewer #2:*

I have reviewed the revised manuscript. For the most part, I believe the authors have addressed concerns raised in the first review. My major outstanding concern is that the 1-tailed tests in Figure 4 are still not appropriate. In panels C-F the authors report neurons with significant effects in directions opposite of their a-priori hypothesis, therefore they are in fact testing both sides of the distribution, but using a 1-sided significance threshold. The authors have the relevant 2-tailed stats and figures in the supplement, and should use these in the main paper.

I think the addition of alternative models has strengthened their results. However, in the table of model parameters, it says that models 1 and 2 have 1 and 2 free parameters respectively. I'm not sure this is correct. For example, model 1 fits a and b, and in the calculation of AICs, the intercept should be included as a parameter, so there should be 2. This shouldn't change the outcome, but stands out as a potential error.

On this note, it seems odd to include the constant, b, in these models at all, when FRs of each neuron are being directly fit from their own FRs. In other words, the underlying hypothesis is that a neuron's FR in the double-cue condition is some mixture of its own FRs in the two single-cue conditions – not that mixture with a constant offset. So why include the constant?

*Reviewer #3:*

The manuscript was significantly improved. However, a couple of points raised in the first review require additional clarification. Numbers refer to those in the first review.

3a) Apropos the idea that the attentional modulation observed in this study represented a "worst-case scenario": In the initial review, it was noticed that one could make the opposite speculation. In a choice setting, which requires using values computed in OFC, monkeys would have considered both options more evenly. Hence, contrary to the authors assertion, had the monkeys been engaged in a choice task, the attentional modulation might have been smaller, not larger. The authors did not take this issue at heart, but they should. Let me note that this is more than a simple discussion point: As the authors make clear in their response to point (1), the worst-case scenario argument is part of why they think the small effects described in this paper are real. Again, I don't see any a-priori reason to think that in a choice setting the attentional modulation would be stronger. If this claim is so important, and if it relies on unpublished data, the data should be described in this paper and we should be given the opportunity to review the evidence. If this claim is not so important, the authors should drop it here and make it elsewhere, when they present the appropriate evidence. In the latter case, they might want to provide a different response to point (1).

3c) If I understand correctly, the authors speculate that offer value cells provide an indirect input to the decision circuit, which operates with working memory mechanisms. However, the fact that offer value cells exist (i.e., the fact that in choice settings some neurons are associated to individual offers), and the fact that noise correlations between them are small implies that these cells are not modulated by attention in the sense described in this manuscript. Do the authors agree on this point? This was the sense of my previous comment, not whether the contribution of offer value cells to the decision is direct or indirect.

---

## [Author Response]

Main comments:1) All reviewers were concerned about the weak neural effects. It is particularly important that rigorous statistical methods are employed to determine whether the effects are real. For example, all reviewers were in agreement that one-tailed tests could not be justified. In addition, there was concern that the effects in Figure 4 are apparent before the stimulus was rotated. This might potentially be a smoothing artifact, but the smoothing window is not reported, making it difficult to determine whether this is indeed the case.

As we mentioned in the Discussion, the passive viewing task provides a worst-scenario estimation of the neural effects (please also refer to #3). Although the size of the effects is small, they are significant and consistent across the population.

The data points in the PSTH curves in the original Figure 4 were made by calculating the mean firing rate in a sliding window of 100 ms. Thus, a smoothing artifact may partly explain the spurious rotation effects before its onset in the plot. To alleviate the artifact, we replaced Figure 4 with the same analysis but now with a sliding window size of 50 ms. The apparent difference between the two curves in Figure 4 before the rotation onset is not significant, and can be probably attributed to noise.

We also believe the choice of one-tailed analyses is justified, because we are testing against the hypotheses that the visual perturbation of the lower value cue would decrease the responses for the positively tuned neurons and increase the responses for the negatively tuned neurons. Nevertheless, we are providing the statistics based on two-tailed analyses in the Supplementary file 1. We also did the analyses in Figure 4 again with the two-tailed analyses and plotted the results in Figure 4—figure supplement 5. The general trend holds and the same conclusion can be drawn based on the new analyses.

2) Reviewers were not convinced by the modeling component of the manuscript. One possibility is that the neural effects arise as a consequence of dividing attention to the two stimuli when the lower value is perturbed. Thinking of the effect as resulting from divided attention, the firing rate in the rotation trials would simply equal the (possibly weighted) average between those associated with the two cues. This account would not need any free parameter (or at most one capturing the weights) and it is very consistent with the data. This would be a simpler model than the selective attention model with four free parameters. The authors should do a formal model comparison with their chosen model against these potentially simpler models, to show that the more complex model is a better predictor of neural activity.

To address the referees’ concerns, we have simplified the old model and now use a slightly different one, which requires only 3 free parameters instead of 4. That is because we now only model the condition in which the rotation is applied to the lower value cue. This simplification does not undermine our conclusions, because the analyses in Figure 3 showed that the attention did not shift when the higher value cue was rotated.

In addition, we tested the new model against two simpler models. The results show that the model that takes into account that the attention distribution should correlate with the value difference between two cues explains the data best.

Please see the more detailed explanation of the new model and the comparison against two other models in the main text.

3) Several points made in the Discussion were unconvincing or problematic.a) "Although the observed attentional modulation of the neural responses in the OFC in this study was relatively modest, it represented a worst-case scenario." One could make the opposite claim: If the behavioral task had required using values computed or represented in the OFC, monkeys would have considered both options more evenly. In other words, contrary to the authors speculation, the attentional modulation might have been smaller had the monkeys been engaged in a choice task.

The point about “worst-case scenario” means that the manipulation of attention is very subtle in this case, as the monkeys had little incentive to pay attention to the lower value stimulus, even with visual perturbation. Thus, the modest attentional modulation could be due to the fact the attentional manipulation is very subtle. This argument is further corroborated by our follow-up study, in which we show an almost complete shift of OFC responses when the monkeys were tested with a Posner style attention task. The preliminary results were reported in a poster at this year’s SfN:

Zhang Z., Xie Y., Yang T., (2017) Top-down attention modulates activity of value-encoding orbitofrontal neurons. Society for Neuroscience Abstract. 45:249.18

As the referee pointed out, in a choice task, the monkey would have to consider both options and we would have a good control of where the attention is. This is precisely why we avoided choice tasks in the first place.

We need to emphasize that making choices is not the key to have a more robust attention control. As the result, we have changed the sentence about the choice task to “We speculate that the attentional modulation would be larger had the monkeys been engaged in a task that demands focused attention”.

b) The authors discuss the neuronal effects described here as "explained by attention". However, the effects recorded here differ in fundamental ways from the attentional effects often described in sensory regions. Normally, the effect of attention on the activity of a neuron is understood as a gain modulation on a sensory response. For example, neurons in visual areas have receptive fields; if the animal pays attention to a particular location, the visual responses of neurons whose RF coincides with that location are enhanced. Concurrently, the visual responses of neurons with different RFs are somewhat reduced. In contrast, OFC cells are not spatially selective – that is, they don't have a receptive field at all. (This is fundamental difference between OFC and V4, MT, etc. Hence, the first paragraph of the subsection “OFC and the visual system” is incorrect.) Thus the neuronal effects found here may have to do more with mental focus than with attention as defined in the context of sensory systems, even though at the behavioral level the effects are driven by a shift of attention.

We actually would like to argue that this experiment may be most comparable to the studies of the visual areas in which multiple stimuli were put in the same receptive field of a neuron. As OFC neurons are not typically spatially selective, one can consider them to have an all-inclusive receptive field. Then we can make meaningful comparisons between the two.

We have modified the Discussion section that compared our results to the visual system studies and hopefully have made our point clearer.

c) The authors re-interpret previous findings of neurons coding the chosen value as neurons coding the value the animal was focusing on. This may be an acceptable interpretation. Then they write "Our results suggest that the OFC encodes the value information one item at a time even when multiple items are presented" and "our results suggest that value-based decision making is a sequential process in which the value of each option is evaluated one at a time, guided by attention". These statements are completely speculative and they seem at odds with previous results. Along with neurons coding the chosen value, several studies found neurons coding the value of individual offers, or offer values. Furthermore, these cells have low noise correlations and low choice probabilities (Conen and Padoa-Schioppa, 2015). If trial-by-trial fluctuations in the activity of these neurons and behavioral choice variability were both driven by attention, one would expect much higher measures for both noise correlation and for choice probability. Thus it might be the case that only some of the value coding neurons in OFC are subject to the effects described here.

We have further evidence from our recent experiments to support our claim at the time of writing. Yet we admit that in the current manuscript, the claim may appear to be too strong. We have made modifications in the Discussion to say “Our results are consistent with the hypothesis…”.

Regarding the low choice probabilities among these OFC neurons, we speculate it could be because choices are made not directly based on the activity of these neurons. In our hypothesis, the decision making is a sequential process in which the OFC evaluates each option. The value information of previously evaluated option has to be held in a different “working memory” brain area for the purpose of decision making. Thus, the OFC’s contribution is indirect and its neurons’ choice probability may be low.

[Editors' note: further revisions were requested prior to acceptance, as described below.]

Reviewer #1:My thanks to the authors for addressing many of the concerns. While this manuscript is improved, there are still very serious issues that have not been resolved.The most significant issues concern Figure 4 which shows how neural activity changes around the time of cue rotation. In brief: there is a lack of explanation for early onset of the effects; there is still a lack of statistical rigor in the significance testing; and there is a glaring data discrepancy between the primary and supplemental figures. Given these issues, I don't have confidence in these data, which are the main results of the paper. All three of these deficiencies – the first being the most critical – must be addressed. In more detail:1) The original version of Figure 4 showed spurious neural activity changes before rotation. While the pre-rotation effects have largely disappeared, there are still neural effects during rotation, and these are still too early to be caused by the rotation. The authors must offer some plausible explanation for this. While they claim it could be due to noise, this is unlikely, as the effect is evident in both monkeys (Figure 4—figure supplement 1), for both positive and negative coding neurons.To elaborate: In Figure 4, the latency to a significant effect is 30ms, which the authors attribute to a "bottom-up" process. However, the initial response to cue onset (presumably also a bottom-up process) occurs at a much slower latency of approximately 70-80ms. The situation is even worse in Figure 4, where the effects are significant at the onset of cue rotation at t=200ms. This first significant bin spans from t=175 to t=225ms, meaning that it includes data from the first 25ms after rotation begins. This is far too little time for rotation-induced attentional effects to appear, given that minimum visual response latencies within the lateral geniculate nucleus are approximately 20ms (Schroeder CE et al. Brain Res. 1989 Jan 16;477(1-2):183-95.).If this were my data, I would strongly suspect an artifact of some kind. For example, if the trial conditions were insufficiently randomized, it is possible that the monkeys could anticipate which stimulus was going to be rotated on some trials, so that they could shift attention before the rotation begins.

We carefully reviewed the data and the analyses for possible explanations of the spurious attentional modulation before the visual perturbation onset. One potential explanation is that the monkeys might be able to predict the perturbation before its onset. Because all trials were arranged in blocks, in principle the monkeys might be able to guess what the remaining trials were towards the end of a block. To test this possibility, we did the analyses in Figure 4 again, but with only the first 50 completed trials from each block (block size = 85 trials) with the value of perturbed and non-perturbed cues balanced (see Materials and methods). In this case, we no longer observe a significant difference between the two curves before the perturbation onset. The results can be found in the new supplementary figure (Figure 4—figure supplement 3,Figure 4—figure supplement 4) and we have added discussion of this issue in the main text.

Considering this potential confounder, we no longer try to estimate the onset of attentional modulation based on Figure 4. This does not affect the main conclusion, which is that the OFC neuronal responses were modulated when the monkeys covertly shifted their attention.

2) In Figure 4 a one-tailed test is not justified. If this were a follow-on study looking to confirm a previously documented effect, then the authors could claim a priori justification for using a one-tailed test. But this study is an investigation of a novel phenomenon, and so the threshold for rejecting the null hypothesis must be suitably strong. While Figure 4—figure supplement 5 uses a two-tailed test, it does not, according to the legend, correct for multiple comparisons. For this analysis to be credible, it must be performed with two-tailed tests, with some form of p-value or FDR correction.

We apologize for the confusion here. Original Figure 4—figure supplement 5 did correct for multiple comparisons. We have rewritten the legend as follows, to indicate this fact: “… (Figure 4—figure supplement 5) plotted in the same convention as Figure 4…”.

In the new revision, we have decided to take the advice of the reviewers and only present the two-tailed analyses. We have removed the one-tailed analyses and figures. New Figure 4 is the original Figure 4—figure supplement 5. Multiple comparison corrections are still included. We have updated all the manuscript and the figures accordingly.

3) In Figure 4 there is a discrepancy between the main and supplemental figures. The same data are supposedly plotted in Figure 4 and Figure 4—figure supplement 5, the only difference being the use of two-tailed tests to draw the significance indicators at the top of the plot. However, the means are clearly different between the two figures. For example, in Figure 4, the red and blue lines overlap at ~850ms, but in Figure 4—figure supplement 5 they do not. These are clearly different data.

Both the original Figure 4 and Figure 4—figure supplement 5 included only neurons that show significant attentional modulation, i.e. the data points with solid colors in the original Figure 4 and Figure 4—figure supplement 5, respectively. Because the selection criteria are different between Figure 4 and Figure 4—figure supplement 5 (1-tailed vs. 2-tailed analyses), they included different sets of neurons. Thus, the curves appeared slightly different.

Again, we now only show the two-tailed analyses. The new Figure 4 is the original Figure 4—figure supplement 5. The one-tailed analysis figures have been removed.

Reviewer #2:I have reviewed the revised manuscript. For the most part, I believe the authors have addressed concerns raised in the first review. My major outstanding concern is that the 1-tailed tests in Figure 4 are still not appropriate. In panels C-F the authors report neurons with significant effects in directions opposite of their a-priori hypothesis, therefore they are in fact testing both sides of the distribution, but using a 1-sided significance threshold. The authors have the relevant 2-tailed stats and figures in the supplement, and should use these in the main paper.

We have taken the advice and now present only the 2-tailed analyses in the main paper.

I think the addition of alternative models has strengthened their results. However, in the table of model parameters, it says that models 1 and 2 have 1 and 2 free parameters respectively. I'm not sure this is correct. For example, model 1 fits a and b, and in the calculation of AICs, the intercept should be included as a parameter, so there should be 2. This shouldn't change the outcome, but stands out as a potential error.

We apologize for the confusion. The number of model parameters here refers to the parameters for each individual neuron. In model 1, we fit each neuron with a unique *b*, and *a* is the best-fitting parameter for the population, which is the same for all neurons. Therefore, model 1 has only 1 free parameter for each neuron. In contrast, in model 2, both *a* and *b* are allowed to vary between neurons. We have clarified this point in the main text.

On this note, it seems odd to include the constant, b, in these models at all, when FRs of each neuron are being directly fit from their own FRs. In other words, the underlying hypothesis is that a neuron's FR in the double-cue condition is some mixture of its own FRs in the two single-cue conditions – not that mixture with a constant offset. So why include the constant?

We agree with the reviewer here, and the distribution of *b* indeed centers around 0. However, removing *b* from model 3 leads to a significant decrease of the model performance, reducing R^2^ from 0.8214 to 0.6557. Adding *b* allows the model to account for other factors that might affect neurons’ responses between different sessions.

Reviewer #3:The manuscript was significantly improved. However, a couple of points raised in the first review require additional clarification. Numbers refer to those in the first review.3a) Apropos the idea that the attentional modulation observed in this study represented a "worst-case scenario": In the initial review, it was noticed that one could make the opposite speculation. In a choice setting, which requires using values computed in OFC, monkeys would have considered both options more evenly. Hence, contrary to the authors assertion, had the monkeys been engaged in a choice task, the attentional modulation might have been smaller, not larger. The authors did not take this issue at heart, but they should. Let me note that this is more than a simple discussion point: As the authors make clear in their response to point (1), the worst-case scenario argument is part of why they think the small effects described in this paper are real. Again, I don't see any a-priori reason to think that in a choice setting the attentional modulation would be stronger. If this claim is so important, and if it relies on unpublished data, the data should be described in this paper and we should be given the opportunity to review the evidence. If this claim is not so important, the authors should drop it here and make it elsewhere, when they present the appropriate evidence. In the latter case, they might want to provide a different response to point (1).

We take the reviewer’s point. What we meant to say is that the attentional modulation may be much larger in tasks in which animals have to actively focus their attention at one of the stimuli. It does not exclude the possibility that in certain tasks, the animals may distribute its attention evenly between the options, which is probably what the reviewer is referring to, but not what we were talking about.

Now we have rephrased the paragraph to reflect the reviewer’s point: “Although the observed attentional modulation of the neural responses in the OFC in this study was relatively modest, it represented a scenario in which the animals were not actively directing their attention toward a stimulus.”

3c) If I understand correctly, the authors speculate that offer value cells provide an indirect input to the decision circuit, which operates with working memory mechanisms. However, the fact that offer value cells exist (i.e., the fact that in choice settings some neurons are associated to individual offers), and the fact that noise correlations between them are small implies that these cells are not modulated by attention in the sense described in this manuscript. Do the authors agree on this point? This was the sense of my previous comment, not whether the contribution of offer value cells to the decision is direct or indirect.

The choice experiments such as in Padoa-Schioppa and Assad, 2006 used different juice types as rewards. The finding of “offer-value” cells that are juice-type specific is potentially compatible with our results, although we don’t know for sure. The current study used only one juice type. It would be interesting to extend the current study and directly test our hypothesis by looking at how attending to different juice type would modulate OFC neurons’ responses.

The small noise correlation observed in the OFC is not incompatible with our hypothesis. First, small noise correlation has also been reported in other prefrontal areas. For example, Qi and Constandinidis (2012) reported that in monkeys performing a saccade task, the noise correlation between pairs of dlPFC neurons was 0.08 when the distance was 0.2-0.5mm and 0.034 when the distance was 0.5-1 mm. This is in the range of the results from Conen and Padoa-Schioppa, 2015. Note that dlPFC neurons are known to be heavily modulated by attention. Second, it has been reported that attention decreases noise correlation in the sensory cortex (Cohen and Maunsell, 2009). Here, we hypothesize that the OFC only represents attended stimuli. One may imagine that the small noise correlation between OFC neurons is inherited from the population of early sensory neurons that represent the attended the stimuli.

These are interesting discussion points, and we have included them in the manuscript. We admit that our hypothesis still requires future studies for support.